# A Warning Call for Fertility Preservation Methods for Women Undergoing Gonadotoxic Cancer Treatment

**DOI:** 10.3390/medicina57121340

**Published:** 2021-12-08

**Authors:** Claudia Mehedintu, Francesca Frincu, Andreea Carp-Veliscu, Ramona Barac, Dumitru-Cristinel Badiu, Anca Zgura, Monica Cirstoiu, Elvira Bratila, Mihaela Plotogea

**Affiliations:** 1“Carol Davila” University of Medicine and Pharmacy, 050474 Bucharest, Romania; claudiamehedintu@yahoo.com (C.M.); frincu.francesca@gmail.com (F.F.); andreea_veliscu@yahoo.com (A.C.-V.), ramona.barac@live.com (R.B.); medicanca@gmail.com (A.Z.); dr_cirstoiumonica@yahoo.com (M.C.); elvirabarbulea@gmail.com (E.B.); 2Department of Obstetrics and Gynecology, “Nicolae Malaxa” Clinical Hospital, 022441 Bucharest, Romania; nicole_plotogea@yahoo.co.uk

**Keywords:** fertility preservation, ovarian reserve, gonadotoxic treatment, cancer

## Abstract

Malignant hematological conditions have recognized an increased incidence and require aggressive treatments. Targeted chemotherapy, accompanied or not by radiotherapy, raises the chance of defeating the disease, yet cancer protocols often associate long-term gonadal consequences, for instance, diminished or damaged ovarian reserve. The negative effect is directly proportional to the types, doses, time of administration of chemotherapy, and irradiation. Additionally, follicle damage depends on characteristics of the disease and patient, such as age, concomitant diseases, previous gynecological conditions, and ovarian reserve. Patients should be adequately informed when proceeding to gonadotoxic therapies; hence, fertility preservation should be eventually regarded as a first-intention procedure. This procedure is most beneficial when performed before the onset of cancer treatment, with the recommendation for embryos or oocytes’ cryopreservation. If not feasible or acceptable, several options can be available during or after the cancer treatment. Although not approved by medical practice, promising results after in vitro studies increase the chances of future patients to protect their fertility. This review aims to emphasize the mechanism of action and impact of chemotherapy, especially the one proven to be gonadotoxic, upon ovarian reserve and future fertility. Reduced fertility or infertility, as long-term consequences of chemotherapy and, particularly, following bone marrow transplantation, is often associated with a negative impact of recovery, social and personal life, as well as highly decreased quality of life.

## 1. Introduction

Malignant conditions, among them hematological ones, have lately proved an increased incidence among young, reproductive-age humans. Leukemia, lymphoma, breast cancer, sarcoma, cervical cancer, and melanoma are among the most common malignant diseases that occur before the age of forty [1]. Onco-hematological conditions face the most aggressive treatment protocols, such as bone marrow transplants, with terrible consequences over the reproductive tissue [2].

The effect of aggressive cancer treatment upon reproductive tissue is often not considered when facing a life-threatening condition. Unfortunately, the destruction could be extensive and permanent due to the nonrenewable characteristics of ovarian reserve. If not permanent, temporary infertility occurs with high incidence [3]. The decrease in follicle pool impacts the window of opportunity for the patient to procreate. Pregnancy, as well as fertility preservation procedures, are not recommended within the first two years following aggressive chemotherapy treatment, as bone marrow transplantation. For a patient with a decreased ovarian reserve, this amount of time reduces, even more, the opportunity to use the remaining follicles. Aside from the follicle impact, damage to the genital tract is also common. Additional negative impact reflects on both general genital function and pregnancy outcome [4,5].

Considering that chemotherapy, regardless of association with radiation, frequently leads to the massive or total destruction of the follicular pool, fertility preservation should stand nowadays on the priority list before, during, or following the cancer protocol [6]. The gold standard in preserving fertility is embryo cryopreservation. At least oocyte preservation should be performed before the onset of cancer treatment, though the time frame before gonadotoxic therapy [7]. The patient’s condition and general health also represent a challenge for the medical team. Various drugs have been proposed to be incorporated during cancer treatment to protect the ovarian reserve, but research is still required before human use [8,9].

This review aims to raise the awareness of the multidisciplinary medical team regarding fertility consequences for cancer patients when administrating specific and proven gonadotoxic treatment. The negative impact upon the genital tract is difficult to overcome following cancer treatment. Chemotherapy and irradiation are associated with permanent damage to the genital system and follicle pool [10]. Mechanism of action is profound and drug-specific, sometimes even challenging to both understand and prevent [5,11].

The psychological impact of childbearing interruption is often found to have deep and extensive negative consequences. It can impair disease recovery, social reinsertion, personal and family life, as well as a great decrease in quality of life. Women tend to report a better acceptance of procreation issues when prior informed by the medical team, rising the importance of proper medical care before the treatment onset [9,12].

## 2. Materials and Methods

A research of the literature was conducted in the databases of PubMed and EMBASE to select full-length articles published in peer-reviewed journals up to October 2021. Both mechanisms of action related to gonadotoxic chemotherapy and irradiation, as well as fertility preservation protocols, were analyzed. The purpose is to highlight the impact on ovarian tissue and the long-term consequences upon the genital tract.

The keywords included in the search strategy were fertility preservation, ovarian reserve, gonadotoxic treatment, radiotherapy and cancer.

## 3. Results

### 3.1. The Effect of Cancer Treatment on Genital Tract

The ovarian tissue may be highly responsive to different types of medication. The follicular pool is predetermined, limited, and non-renewable and should be protected against external adverse effects. Environmental factors impact fertility and genital function overall. Cancer treatment affects both, directly and indirectly, the ovarian tissue, endocrine, and fertility function [13,14].

### 3.2. Ovarian and Follicular Pool Characteristics

To have a better understanding of cancer treatment’s impact on gonads, it is important to view the genital tract as a whole and to acknowledge the complexity of the ovarian tissue and its functions. Histologically, the ovarian cortical area holds the follicles, both growing and dormant. The surrounding stroma from the cortex is a mage of fibroblasts, which are highly sensitive to hormonal secretion, unlike the fibroblasts from the rest of the body. The medulla, the internal area of the ovary, withholds the vascular network [15,16].

Follicles form from the primordial germ cells in the intrauterine fetal period. The 1000 germ cells migrate to the future gonadal area, enter initial rapid mitosis, followed by the first meiosis and arrest in a few steps of prophase. Forming the initial 5 to 7 million follicles after the 5th month of intrauterine life, only 1 to 2 million are available after birth because of a process of apoptosis and atresia [17]. The rest of the oocytes will be surrounded by a layer of somatic pre-granulosa cells, thus forming primordial follicles. Only high-quality oocytes further develop and fertilize during adult life [18]. Bidirectional communication and signaling between the oocytes and the outside granulosa cells, as well as the regulation of the primordial follicle assembly in the fetal period, are responsible for this selective mechanism, as demonstrated by recent studies. The follicular atresia continues throughout childhood and adult life, allowing only 400–500 follicles to transform into mature follicles and ovulate [19].

Folliculogenesis encounters in the ovarian cortex, and constantly there is a very sensible balance between dormant and growing follicles. Among the central processes that define the mechanism are the recruitment and activation of the primordial follicles, the development and reaching the preantral stage, the selection over the antral stage, and then ovulation or atresia [20]. Folliculogenesis divides into two parts based on hormonal sensitivity. The preantral part is gonadotropin-independent and characterized by oocyte growth and differentiation under the stimulus of local growth factors, cellular and subcellular mechanisms. The second, gonadotropin-dependent antral phase, is characterized by the rapid growth of the follicles under the feedback mechanism of FSH (Follicle-stimulating hormone), LH (luteinizing hormone), and gonadotropins. The onset of folliculogenesis starts with the recruitment of a primordial follicle orientated towards growth and differentiation [21]. This process takes about ten menstrual cycles or approximately 290 days for the follicle to reach the second stage and one year until the ovulatory phase. Technically, the ovarian function should be evaluated one year after the cessation of gonadotoxic treatment, meaning the necessary time for the growing follicle pool to be completely renewed [22].

The recruitment of dormant follicle is under the influence of very sensible mechanisms [23]. The purpose is to keep the ovarian reserve with primordial follicle sparing and a perfect balance between inactive and active follicles. Once the percentage of growing follicles decreases, the recruitment is activated [24]. The dormant follicles are kept inactive by subcellular pathways that inhibit the activation process. The inhibiting mechanism can be inactivated indirectly by the lack of AMH (anti-müllerian hormone) secreting follicles, thus depending on serum AMH [25]. Additionally, several medications, such as alkylating agents commonly used in cancer treatment, directly inactivate the pathways, so the recruitment is accelerated [26,27,28].

A few pathways were described, such as PI3K/PTEN/Akt, mTOR, and FOXO3A. Phosphatidylinositol-3-kinase (PI3K), phosphatase and tensin homolog (PTEN), protein kinase B (Atk), mammalian target of rapamycin (mTOR) [29,30], and forkhead box class O 3a (FOXO3A) are among the major pathways and signaling methods involved in cell growth, proliferation, differentiation, metabolism, as well as apoptosis and stress management [23,31]. In the primordial follicle pool, those pathways are responsible for recruitment, growth, keeping a high proportion of the inactive follicle, and contributing to follicle growth, survival, development, and response to DNA aggression and damage [32]. The actions and functions of these pathways upon normal cells activity and cancer development were intensely studied. Besides keeping the homeostasis of the follicle, targeting those inhibitory pathways [33] is the key to protecting the ovary during gonadotoxic chemotherapy, ovarian aging, and ovarian tissue transplantation [29,34].

During a gonadotoxic treatment, the negative effect acts on both variants. At first, systemic administrated drugs suppress the growing follicles, leading to a decrease in serum AMH. Following this, the rapid activation of follicular recruitment restores the AMH serum concentration and balances the two types of follicular pools [35]. If Cyclophosphamide, an alkylating agent, is administrated, the inhibiting mechanisms are directly inactivated and the activation of dormant follicle accelerates, thus exposing more follicles at risk. If the exposure to gonadotoxic therapy is prolonged, more follicles destroy, and eventually, the burnout effect takes over the ovary, leaving behind no ovarian reserve [36,37].

### 3.3. Cellular Apoptosis 

The central mechanism responsible for cellular death is apoptosis, also known as the process of programmed cell death. The majority of external factors impact the structure of cell DNA and lead to chain ruptures. The altered cells will be removed via apoptosis, while the damage cannot be restored [28]. This direct action is the effect of chemotherapy and irradiation on ovarian tissue. Active and growing follicles are at high risk during aggressive treatments, with massive cell alteration and induced apoptosis, leading to temporary amenorrhea [38]. The longer the treatment, the greater the follicles exposure and subsequent death, followed by a rapid decrease in serum estrogen, AMH, and increased FSH [39,40].

### 3.4. Acute Vascular Toxicity

Stromal cell damage is one of the systemic consequences of administrating chemotherapy. Heterogeneous alterations upon blood vessels, reduction in blood flow and volume, vascular spasm, architecture disruptions with vessels disintegration, as well as fibrosis would consequently appear [13,14]. Vascular toxicity on ovarian tissue mostly affects the cortical region and is followed by acute follicular ischemia of the growing follicles. Primordial follicles were thought to be protected against direct vascular toxicity because they do not depend on blood supply like the growing ones [41]. Recent studies confirmed that the follicular pool is also sensitive to this destructive mechanism [42].

### 3.5. Ovarian Burnout

Ovarian burnout represents the best example of indirect damage associated with cancer treatment. The ovarian follicular pool is in a delicate and perfect homeostasis state. Both internal and external factors contribute to a dormant state that protects the ovarian reserve. The activation of the primordial follicle is very complex [33]. Researchers have tried to cover the underlying mechanism that leads to massive or even total follicle depletion during continuous and prolonged chemotherapy [43]. From the dormant state, a primordial follicle activates when needing to take part in the growing pool. External factors such as AMH serum concentration, known to be secreted by small growing follicles, also contribute [37]. The dormant reserve is not sensitive to variations of FSH, LH, or serum estradiol concentrations but is responsive to subcellular pathways that keep them in a non-active state. The activation of the PI3K/PTEN/Akt pathway is crucial for the oocyte reserve [44]. This pathway is inhibited by the AMH concentrations, with subsequent activation of the mechanism in the absence of small AMH secreting follicles [45]. This stimulation process followed by continuous activation of the dormant follicles is proportional to the constant growing follicle destructions due to prolonged chemotherapy. Decreased AMH serum concentration provides this feedback response. Similarly, some chemotherapy medications, such as alkylating agents, directly activate the PI3K/PTEN/Akt pathway followed by follicle awakening and more growing follicles [33,37]. Those would be sensitive to the direct actions of cancer treatment, with permanent damage and a subsequent need to activate dormant follicles to restore the primordial/growing follicle balance [32]. Both mechanisms are dose-dependent and cumulative, eventually leading to primordial follicle complete depletion and related ovarian failure [46].

### 3.6. Oxidative Stress

It is well known and previously confirmed that antioxidants play an important role in follicles’ survival. Cellular oxidative stress is related to exposure to Cyclophosphamide (an alkylating agent commonly used to treat onco-hematological conditions) [43]. It is present both within granulosa cells and follicles. The mechanism is associated with increased oxygen-derived free radicals and reduced antioxidants, leading to apoptosis of the damaged cell [5,13,47].

### 3.7. Irradiation 

Whether as a hematopoietic stem cell transplantation regimen (HSCT) or as a directed and limited procedure on the genital tract, total body irradiation associates devastating effects upon ovarian tissue [48]. Primordial follicles are highly sensitive to ionizing radiations. Most of them cannot restore the irradiation-induced damage and will undergo phagocytosis [49]. The effect is related to cellular development, and the more mature, the greater the damage induced. The remaining functional ovarian reserve depends on age, dose, and number of radiation procedures. The cutoff limit for permanent damage is 10 Gy, commonly reached during transplant protocol [50]. A dose of 4 Gy induces damage in almost half of the ovarian reserve, while sterility establishes at 20 Gy in young women and only 6 Gy in over 40 patients. Even though the rate of destruction depends on age, the regimen received after the age of 25 is highly toxic regardless of dose or fractioning [47,51].

Aside from follicle impact, irradiation also damages the uterus, with reduced distension of the cavity and thinning of the endometrium. Those alterations are associated with increased abortion rate, premature deliveries, and low for gestational age fetuses [52,53].

Other endocrine glands are also affected due to total body irradiation. Decreases in gonadotropic hormones release, hyperprolactinemia, secondary infertility, and abnormal steroid hormones secretion are also consequences of irradiation of the hypothalamic and pituitary areas [54,55].

### 3.8. Fertility Preservation Guidelines

Anticancer therapies may have profound consequences upon the genital system through chemotherapy, associated or not with irradiation exposure [56]. Secondary amenorrhea and infertility, either temporary or permanent, are the most important concerns regarding cancer treatment effects [13]. The degree of destruction depends on the age, medical condition, treatment protocol, and required procedures, including types and doses of agents, irradiation, and the medical profile, prior and concomitant medical history [39,57]. The remaining functional ovarian following gonadotoxic cancer treatment is hard to measure, especially regarding future fertility and oocytes quality. Each patient should be extensively evaluated and correctly informed before starting cancer treatment [12]. Doctors must advise every woman about adverse and destructive effects on genital, endocrine, and reproductive functions [58]. When available and requested by patients, fertility preservation procedures should be presented and performed accordingly. The oncofertility specialist should evaluate and decide among types of fertility preservation procedures, as well as the suitable ovarian stimulation protocol. Ultrasound characteristics, age, medical condition and the menstrual cycle phase of the patient will determine the type and duration of the fertility preservation protocol [59]. Additionally, future fertility and procreation opportunity are closely related to the retrieved oocytes, both number and quality. The best results are achieved when the procedure is performed before cancer treatment, but in some cases, this is not possible because of the disease, general conditions, or following risks [60,61].

### 3.9. Fertility Preservation before Cancer Treatment

Though aggressive cancer treatment has proved its benefits in decreasing overall cancer-associated mortality and increasing the survival rate, the ovarian tissue long-term consequences should be just as considered as malignancy’s treatment. Potential fertility issues related to cancer treatment should be explained to patients, especially when the required therapy is known to have a major impact on reproductive function [48]. Preserving fertility in the context of cancer treatment has been proven to be a challenge for oncofertility specialists. It can be performed either by preserving oocytes, embryos, ovarian tissue, or transposition of the ovary. The available and suitable procedures depend on various external and internal factors, as well as financial and logistic [6,62].

The gold standard in fertility preservation before cancer treatment is embryos cryopreservation [8,9]. If embryos are not an option, mature oocytes should be targeted. Consent from partners and single patients are among the limitations regarding the possibility of embryos cryopreservation [63]. Ovarian biomarkers, such as FSH, LH, AMH, and AFC (antral follicle count), evaluate the ovarian reserve [64,65,66]. The time frame before cancer treatment initiation is very narrow; therefore, the ovarian phase should be considered when choosing a protocol [67]. Nowadays, two weeks are sufficient to stimulate and retrieve oocytes regardless of the ovarian cycle phase. At best for future fertility is to obtain and collect a high number with good quality oocytes from the patient. Standard protocols depend on spontaneous menstruations and include GnRH (gonadotropin-releasing hormone) agonists to lower the hyperstimulation syndrome, but commonly used stimulation protocols in the context of cancer emergency preservation now use GnRH antagonists to reduce procedure duration [68]. Doses and type of stimulation medication are related to age and ovarian reserve and are interdependent to increase the number of resulting oocytes [69]. Following the procedure, oocytes could be frozen or fertilized with embryos cryopreservation [70]. If no mature oocytes can be retrieved, immature oocytes could be obtained, and the procedure can be performed regardless of the menstrual cycle. They will be subjected to in-vitro maturation procedures, as it happens when given hCG to small follicles or in the case of prepubertal patients. Unfortunately, the results are considerably inferior to mature oocytes, given that the maturation rate is approximately 50–60%, and the fertility rate riches 60–70% [71].

If cancer treatment cannot be delayed in order to stimulate the ovary and retrieve oocytes, the patient can benefit from ovarian tissue cryopreservation [72]. This is the only available option for prepubertal patients that would be exposed to gonadotoxic medication and patients with acute leukemia [73]. Cryopreservation can be performed by harvesting the whole ovary, keeping in mind that the ovarian cortical area contains primordial follicles. Harvesting will also include immature oocytes, later to be subject to invitro maturation procedures. For functionality preservation, both the technique and the graft sizes are important. The actual recommendations are to slow freeze and fast unfreeze small ovarian tissue slices, either (8–10) mm× 5 mm or 2 mm × 2 mm [9,74]. This procedure has its limitations, such as the low quality of the ovarian tissue but also relapses of cancer after introducing the graft in the ovarian area, especially when the initial recommendation was acute leukemia [75,76]. Studies show that the ovarian graft function and lifespan restore after approximately 4–6 months. The decrease in serum FSH and increased estradiol confirm this. Regarding conception following the procedure, the positive result with pregnancies rises up to 50% after spontaneous menstruation onset [74]. When reintroducing graft is not an option, an artificial ovary may represent an alternative solution [77]. The method is promising, but until now, it has been used only in lab mice models. This future procedure could lower the risk of reintroducing malignant cells when auto-transplant the ovarian graft [78,79].

Transposition on the ovary into another place within the patient’s organism has its indications when pelvic irradiation is required [80]. Unfortunately, it does not protect when total body irradiation or systemic gonadotoxic chemotherapy is part of the treatment regimen [75,81].

### 3.10. Fertility Preservation during Cancer Treatment

Malignant hematological conditions often require immediate treatment in order to save the patient’s life, leaving behind fertility preservation [82]. The two weeks needed for cryopreservation of oocytes or embryos could sometimes not be available for many patients, even though there is the urgency of treatment onset or the characteristics of the disease, as is the case for leukemia. In the case of acute leukemia, the only available option is ovarian tissue cryopreservation. Regarding this medical condition, it is known that both delaying the treatment and autotransplantation of the ovarian graft is not often an option [56]. In all those cases, once the treatment has already started, options narrow especially because of the negative effect of chemotherapy on growing follicles. Many studies and researches attest that fertility protection can be achieved during gonadotoxic treatments [83].

GnRH is probably the most used and known, but the results are not as expected. It is the first drug to be used in order to preserve fertility during cancer treatment, and the first studies started in 1981. An animal study confirmed the gonadal protection of GhRH agonists administration during Cyclophosphamide treatment. Further studies also confirmed superior gonadal protection of prepubertal patients compared to adults, though followed by numerous researchers that reported conflicting data, most of them con confirming follicular protection during gonadotoxic cancer treatment. GnRH mechanism of action is not quite known, apparently both direct and indirect on the ovarian function [84]. Their protection depends on the medical condition and type of chemotherapy. Though inducing a menopausal state with low FSH and hypoestrogenemia, we must keep in mind that follicle activation is not dependent on those ovarian biomarkers [85]. There is a beneficial effect related to the menopausal state induced with mild protection of the follicle pool, especially after four weeks of treatment when the AMH has increased by 30% after an initial decrease. One other protective effect related to GnRH administration is the reduction in ovarian perfusion and exposure to chemotherapy [68]. Lowering the blood flow to the ovarian tissue, unfortunately, leads to local ischemia and additional damage [86]. One other inhibiting mechanism associated with GnRH is believed to be upon follicular subcellular activation pathways, such as PI3K/Akt/mTOR [68]. This pathway may somehow prevent ovarian burnout, but clinical experience has not provided yet enough experience [87,88]. Though not proven to be beneficial for the ovarian reserve, GnRH administration protects patients from vaginal bleeding in the context of cancer treatment that often associates thrombocytopenia [89].

Chemotherapy reduces follicle pool through multiple and complex mechanisms, affecting mainly growing follicles and inducing local fibrosis. Given that mature follicles are impossible to be protected, the main concern of the medical staff is to protect the dormant reserve. Primordial follicles are activated and recruited using subcellular activation pathways that may depend on the serum AMH, as proven already [90]. Considering that AMH harms those pathways, the lower the serum concentration, the less negative effect on those mechanisms. AMH secreting follicles are damaged by gonadotoxic therapy, and primordial follicles are recruited to keep the active/dormant follicle balance [32]. Researchers have developed a method to protect oocyte reserve by administrating recombinant AMH during cancer treatment [91]. This medication has been proven beneficial in offering protection for future fertility in vitro, even in combination with Cyclophosphamide, which is known to be the most aggressive gonadotoxic agent [44]. The in vivo method has its limitations though, mainly because alkylating agents have a direct negative effect on the PI3K pathway, followed by its activation and beginning of the follicle recruitment. Administrating recombinant AMH does not block the process of recruitment by all mechanisms involved, some being active and primordial follicle activated despite serum AMH [86,92]. One other disadvantage is the limited bioavailability of the medication that proved to be undetectable 17 h after administration. Besides many laboratory clinical studies that have proven protection on ovarian reserve, more research is required before introducing the medication in cancer protocol for human subjects [90,91].

The need for fertility preservation during gonadotoxic medication led the scientists to evaluate another treatment for future use [93]. AS101, also known as tricolor ammonium tellurate, is a non-toxic immunomodulator frequently used in cancer treatment [30,40]. Besides other general effects, the drug inhibits PI3K/PTEN/Akt activation pathway and is responsible for follicular activation, keeping the ovarian reserve at a higher level compared to previous medication. Studies have confirmed the protective action of AS101 in association with Cyclophosphamide [45,94]. Additionally, serum concentrations of AMH were measured during the treatment, and they were confirmed to be higher while on AS101 treatment, showing that both growing and dormant follicles are protected. Those in vitro studies still need additional research before human use [37,92].

### 3.11. Fertility Preservation after Cancer Treatment

It is difficult to assess ovarian function following gonadotoxic treatments. There is no standardization regarding ovarian insufficiency or premature ovarian failure following cessation of cancer treatment. Various clinical trials have tried to create an evaluation method in order to predict the recovery or premature failure of endocrine ovarian function, as well as fertility [68].

The common ovarian biomarkers, such as FSH, estradiol, AMH, and ultrasound follicle count (AFC), can only superficially orientate us regarding present ovarian function [95]. Age is probably one of the most important factors in terms of future fertility because the younger the patient, the higher the number and quality of the remaining oocyte [74]. It is also important the moment of evaluation for both paracrine biomarkers and ultrasound findings. The ovarian function should be evaluated after at least six months following the cessation of treatment. This is related to the amount of time that is required for a follicle to be activated from the ovarian pool and then to achieve a growing, ultrasound evaluation state. It is considered that 290 days are required for a primordial follicle to become a secondary follicle and one year to reach the ovulatory state. Therefore, no spontaneous menstruation within a year is frequently associated with permanent ovarian damage and induced menopause [39]. All patients with gonadotoxic therapy for cancer, with temporary ovarian damage, experience premature menopause [96]. The difference depends on the number of remaining primordial follicles. The degree of the primordial follicle destruction could predict the time before the onset of menopause, though challenging when the ovarian biomarkers are found to have low serum concentrations. The time frame for fertility is limited, even for patients who restored their menstruations [97]. Fertility is closely related to the remaining ovarian reserve, but post-cancer treatment medication could additionally impact both oocytes quality and the patient’s possibility of procreation. Additionally, we should keep in mind that the best oocytes are primarily to be used, and those with the poorest quality are the ones that remain. When addressing ovarian reserve, AMH serum concentrations can be helpful, but the quality of the remaining follicles is by far the most important factor regarding fertility and future pregnancies [98].

Following aggressive cancer treatment, options are limited and not promising. Oocytes or embryo cryopreservation could be considered for a patient with a small ovarian reserve and not before 12 months following cancer drugs or procedures cessation [99]. The procedure is not similar to women without gonadotoxic medication exposure. [100] An ovary exposed to chemotherapy has a lower response to ovarian stimulation, as well as a lower number and a poor quality of resulting oocytes [75,101]. One other aspect that should be considered is the general damage of the genital system related to cancer treatment exposure. The impact of chemotherapy and irradiation is reflected in the whole genital system. Impair of the vascular system, the architecture, temporary atrophy, especially related to irradiation, additionally impact the number of pregnancies and the outcome, significantly lowering the resulting full-term live births [53]. If poor quality or no oocytes are available, patients could undergo egg donation ‘In vitro’ fertilization (IVF). This is a valid option for women exposed to cancer treatment, even though studies reported that patients rarely appealed to this procedure compared to the normal population [102,103].

## 4. Conclusions

Malignant conditions, especially in young patients, require aggressive, sometimes gonadotoxic treatments to save lives. Specific cancer therapies are often associated with long-term consequences on many organs and systems, including the genital tract. Mechanism of damage related to follicle pool, both active and dormant, knows several pathways and its characteristic to different types of drugs and cancer procedures. Decrease or destruction of the ovarian reserve most likely occurs as a result of aggressive and gonadotoxic treatment. Future fertility and reproductive life span are among the greatest concerns of the medical team and reproductive specialists. Fertility issues impact disease recovery and quality of life, especially for young nulliparous women. 

When facing a life-threatening medical condition that can associate infertility, women should be properly informed about the probability of genital tract harm and the available and suitable fertility preservation methods. 

## Data Availability

Not applicable.

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
