# Peer review of "A Warning Call for Fertility Preservation Methods for Women Undergoing Gonadotoxic Cancer Treatment"

_medicina, 2021, doi:10.3390/medicina57121340_

Round 1
Reviewer 1 Report
A brief summary
The outline of the present manuscript is to review the impact of gonadotoxic treatment on fertility and preservation methods.
Broad comments
I think many embryologists, medical doctors, and patients are interested in this issue. This review summarizes the impact of gonadotoxic treatment on fertility and preservation methods and the readers can know the current state of techniques how to maintain the reproduction ability against cancer treatment.
Specific comments
Line 130-138: The paragraph is written in bold style.
Line 230-231: fast unfreeze small, either 8-10x5mm or 2x2mm ovarian tissues >> What does this sentence mean? Please check this sentence.
References: Some references are needed to unify according to Instructions for authors.
Reviewer 2 Report
In this review, the authors write a description of the literature on fertility preservation options for female cancer patients that are at risk of future fertility impairment due to gonadotoxic treatments.
This is a fascinating subject, and the authors achieved a somewhat comprehensive summary of gonadotoxic effects on female fertility and fertility preservation strategies. However, the manuscript in its current form, in this reviewers' opinion, does not provide a significant contribution to the literature in the field. Furthermore, there are many aspects in methodology that require the authors' attention in order to meet minimum standards for publication.
Comments to the authors:
- English language requires extensive editing
- The title requires rephrasing/rewriting to reflect more accurately the content of this review
- The aim is not clearly outlined
- The abstract is not structured, please define the aim
- Materials and methods require a more detailed description of how this "systematic" review was actually carried out. If this was not intended to be a systematic review, this needs to be corrected in methods section.
- Results section shows a monograph-style text without a significant in-depth analysis of reviewed literature.
- Conclusion is floating without an anchor to a specific aim, needs to be rewritten according to the aim of the review.
- References and citations are outdated, and cited work does not include enough references showcasing the work of the biggest teams working in this particular subject in the field of human reproduction and female fertility preservation.
Overall, this is a good summary of gonadotoxic treatments and fertility preservation, that provides a nice overview of basic literature on the subject. However, the manuscript does not meet the criteria for a systematic review and requires extensive revision to significantly strengthen it.
Round 2
Reviewer 2 Report
This reviewer believes the authors have addressed most of the raised concerns and the paper has been improved up to a certain degree.
Regardless of the authors' native language, scientific work needs to be impeccably written to properly deliver the message that is intended to convey.
References were not revised, this reviewer strongly suggests the addition of more updated work, reflective of the period of time covered in the materials and methods section. Only 1 reference from 2021 was included, 1 from 2020, 3 from 2019, and so on. This reflects an outdated reference list.
Furthermore, when keywords mentioned in methods are entered in repositories utilized by the authors, there is a much bigger number of papers that could easily be retrieved. According to the authors' inclusion criteria, these papers should have been at least cited or mentioned throughout the manuscript. These issues should be properly addressed by the authors, either by providing a more thorough article inclusion and exclusion criteria or by including more updated papers in the manuscript.
If these issues are properly tackled by the authors, this reviewer believes the paper will be significantly strengthened.
Round 3
Reviewer 2 Report
The manuscript was significantly improved.